# A Sample Preparation Technique Using Biocompatible Composites for Biomedical Applications

**DOI:** 10.3390/molecules24071321

**Published:** 2019-04-03

**Authors:** Huifang Liu, Geun Su Noh, Yange Luan, Zhen Qiao, Bonhan Koo, Yoon Ok Jang, Yong Shin

**Affiliations:** 1Department of Convergence Medicine, Asan Medical Institute of Convergence Science and Technology (AMIST), University of Ulsan College of Medicine, 88 Olympicro-43gil, Songpa-gu, Seoul 05505, Korea; liuhuifang.1229@gmail.com (H.L.); ngs90@hanmail.net (G.S.N.); luanyange@gmail.com (Y.L.); qiaozhen90@hotmail.com (Z.Q.); qhsgksdlek@naver.com (B.K.); jangyo17@daum.net (Y.O.J.); 2Biomedical Engineering Research Center, Asan Institute of Life Sciences, Asan Medical Center, 88 Olympicro-43gil, Songpa-gu, Seoul 05505, Korea

**Keywords:** sample preparation, nanocomposite, pathogenic, enrichment, nucleic acid isolation

## Abstract

Infectious diseases, especially pathogenic infections, are a growing threat to public health worldwide. Since pathogenic bacteria usually exist in complex matrices at very low concentrations, the development of technology for rapid, convenient, and biocompatible sample enrichment is essential for sensitive diagnostics. In this study, a cucurbit[6]uril (CB) supermolecular decorated amine-functionalized diatom (DA) composite was fabricated to support efficient sample enrichment and in situ nucleic acid preparation from enriched pathogens and cells. CB was introduced to enhance the rate and effectiveness of pathogen absorption using the CB–DA composite. This novel CB–DA composite achieved a capture efficiency of approximately 90% at an *Escherichia coli* concentration of 10^6^ CFU/mL within 3 min. Real-time PCR analyses of DNA samples recovered using the CB–DA enrichment system showed a four-fold increase in the early amplification signal strength, and this effective method for capturing nucleic acid might be useful for preparing samples for diagnostic systems.

## 1. Introduction

Pathogenic infections result in diseases caused by toxins released by pathogenic organisms, and such infections are a growing threat to human health and public health worldwide [1,2,3]. Currently, pathogen identification and therapeutic approaches play important roles in controlling infections. However, the traditional gold standard diagnostic method, i.e., culturing and colony counting, is limited by long waiting times (and thus wasted time), as culturing of most clinical bacterial pathogens requires 1–2 days (much longer times are required for several bacterial species), and low efficiency due to contamination and significant experimental error [4,5,6]. Rapid and effective detection technologies are especially and urgently needed for an early-stage diagnosis, at which time there are low concentrations of the target pathogen. Therefore, new technologies based on the use of novel materials for sample preparation and biosensors for highly sensitive detection are emerging.

Among the emerging technologies, ‘enrichment technology’ is playing an increasingly important role in both sample preparation and sensor diagnosis amplification [7,8,9]. Nanomaterials that can be used as nanosorbents and activators for sample preparation have received considerable attention in various applications. A key advantage of such nanomaterials is their usefulness as sorbents. Their large surface area combined with a potential to modify their surfaces with various special reactive groups can increase their chemical affinity to target compounds [10,11,12]. A supermolecular modified diatomaceous earth (DE) composite platform has been used for molecular encapsulation in water treatment and in a broad range of biological systems that require rapid results and high stability [6,11]. In previous studies of functionalized DE composites, the extraordinary three-dimensional porous structure of DE has been shown to supply a massive surface area; furthermore, the well-known process of amino functionalization activates DE, while the addition of the pumpkin-shaped molecule cucurbituril (CB) maximizes its encapsulation performance by improving its dispersion [3,10,13,14,15].

Here, an efficient method for fabricating a cucurbituril-decorated, amine-modified diatom composite (CB–DA) is proposed, and the usefulness of this CB–DA composite for efficient sample enrichment and in situ nucleic acid preparation from pathogens and cells is demonstrated. The well-characterized process of amino functionalization of the surface of DE renders it a universal tool for targeting molecules, and cucurbit[6]uril (CB) has been introduced to enhance the ability of CB–DA composites to rapidly and efficiently adsorb molecules [3,16,17]. We have also verified that modification with 3-aminopropyl-methyl-diethoxysilane (APDMS) is more effective than traditional modification with 3-aminopropyl-triethoxysilane (APTES). The characterization of the surface charge (as represented by the zeta potential) indicates that the charge conferred by APDMS modification is about twice that conferred by APTES modification. Furthermore, the efficiency of the CB–DA composite to enrich for pathogens and cells was examined using three approaches. First, we tested the absorbance of supernatants collected after CB–DA enrichment. The results show that the enrichment efficiency was as high as 90% within 3 min, even at an *Escherichia coli* concentration of 10^6^ CFU/mL. Second, the morphology of the composites precipitated by the CB–DA system showed that numerous eukaryotic cells adhered to the surface of the CB-DA. Third, we also compared the performance of real-time PCR using amplified DNA, samples collected from the CB–DA enrichment system, and DNA extracted via a commercial column system as templates. The results showed approximately a four-fold increase in the early amplification signal. In summary, we have confirmed that this CB–DA composite system can provide improved performance for biosample preparation for early diagnosis.

## 2. Results and Discussion

### 2.1. Design and Principle of the CB–DA Biocompatible Composite

As we reported in our previous study, cucurbituril-based diatom composites (CB–DA) exhibit a strong host–guest interaction that supports molecular encapsulation [11]. An efficient method for fabricating CB–DA to diagnose pathogen is proposed in this study. By examining the DE using scanning electron microscopy (SEM) (JEOL JSM-7500F, Tokyo, Japan) and dynamic light scattering (DLS) (DynaPro NanoStar, Wyatt), we confirmed that the size of the DE particles was well distributed between 10 and 20 μm—an appropriate size range for this project (Figure 1A,B). APTES (3-aminopropyl-triethoxysilane)-modified DE (also known as DA) has an enhanced pathogen enrichment property [11]. Here, we focus on DE surface medication with the similar chemical of diethyl amino polydimethylsiloxane (3-aminopropyl-methyl-diethoxysilane (APDMS)) [12]. The structural formula of APDMS is shown in Figure 1C, and the chain length of the organic amino compound can be seen. A diagram of cucurbit[6]uril [18,19] with a portal diameter of 3.9 Å, a cavity diameter of 5.8 Å, and a height of 9.1 Å is shown in Figure 1D.

### 2.2. Preparation and Characterization of DA

A schematic of the process flow for amino functionalization of diatomaceous earth (DA) with different chemicals is shown in Figure 2A. The diethyls of APDMS are different from the triethyls of APTES for the surface modification of DA. Two points of the bonding site from APDMS is expected to be more efficient at bonding than APTES with three points (Figure 2A). The surface charges of the composites (reported as the zeta potential) were measured to estimate the efficiency of the amino modification (Figure 2B,C). Pure DE exhibited a negative surface charge. Amine groups surrounding the inner and outer surfaces of the DE skeleton can enhance its chemical stability, allow it to be used for extended periods of time, and lead to a robust coating with saline via covalent bond formation. Serial doses of APDMS or APTES, ranging from 50 μL to 400 μL, were added to 500 μL (100 mg/mL) of DE. The zeta potentials shown in Figure 2B,C show that a modification ratio of 1:2 (250 μL into 500 μL of DE) led to a good modification efficiency for both APDMS and APTES. Overall, APDMS modification was more efficient than APTES modification under the same conditions, perhaps due to differences in the contact surface area of the molecules. Modification with APDMS (which contains diethyl groups) requires two sites, while modification with APTES (which contains triethyl groups) requires three sites. To optimize the APDMS modification time (Figure 2D), the modification reaction was monitored from 30 to 180 min. Here, the modification ratio of 1:2 (250 μL into 500 μL DE) was used to measure the optimization of the APDMS modification time. Notably, according to the APTES modification procedure [11], a 120 min incubation time for the modification has been fixed in dosage studies. A modification time of 90 min was chosen for use in the fabrication process. Taken together, these results show that APDMS modification resulted in the presence of two free chains on the DA surface that could be involved in additional linkages.

### 2.3. Preparation and Characterization of CB–DA Biocompatible Composites

A schematic representation of the cucurbituril coating of APDMS-modified DE (CB–DA) is shown in Figure 3A. The two free chains of the APDMS on the substrate are encapsulated in the cavity of the cucurbit[6]uril (CB), which is a key property of this second DE surface modification. Chemical-optical spectrum analysis of the composites was performed to assess the modification status. SEM images were used to assess the morphology of the DE and CB–DA, as shown in Figure 3B. The pores on the DE surface are open, and the pore diameter is less than 100 nm (Figure 3B, top). However, the CB–DA is rougher with blocked pores on the surface (Figure 3B, bottom). To assess the electrostatic properties of the CB–DA in solution, the zeta potentials of the CB–DA were measured (Figure 3C). Due to the uniform size of the DE particles, we ignored the size effect on the surface charge. Notably, the zeta potential of the CB–DA composite was higher than that of the DA, likely reflecting the diverse anchor bindings between the DA and CB, which may include the reported possible anchor linking/ion-dipole interaction between the carbonyl groups of the CB portals. Furthermore, the positively charged amine groups in the CB–DA composite could enhance the absorbency efficiency of the CB–DA conjugate during its interaction with other molecules via enhanced covalent bonding, physical adsorption, electrostatic interaction, and heterogeneous surface binding [11,19,20,21]. In the Fourier-transform infrared (FTIR) spectrum analysis of the composite (Figure 3D), the absorption peak at 1450 cm^−1^ was attributed to asymmetric stretching vibrations in the Si–O–Si bonds, and the peak at 1410 cm^−1^ resulted from the Si–CH_2_ bond (black curve). In addition, the absorption peaks at 3295 and 1180 cm^−1^ can be attributed to Si–OH and C–N bonding, respectively, on the surface of the pure DE. After the surface modification to form DA, the well-defined absorption bands at 1100, 2250, and 2720 cm^−1^ represent C–C–C bonding, C–H bonding, and O–H bonding, respectively. The stretching vibrations at 2850–3000 cm^−1^ from the CH, CH_2_, and CH_3_ groups and that at 2720 cm^−1^ from the aldehyde (C–H) in the CB–DA group (blue curve) verified the presence of a CB–DA supermolecule.

### 2.4. Cell and Pathogen Enrichment Using the Biocompatible Composite

A schematic of the pathogen enrichment process is shown in Figure 4. Electrostatic interactions between the positive surface of the CB–DA and the negative charge of the cell membrane form bridges that facilitate absorption (Figure 4A). The pathogen–composite complex precipitates easily. To assess the enrichment capacity, a UV spectrophotometer (Libra 22 UV) was used to measure the absorbances of the supernatants from the tested pathogen samples containing *E. coli* (10^6^ CFU, 2 mL) after treatment with DA alone or after the CB–DA enrichment process. As shown in Figure 4B, the lower absorbance of the supernatant following CB–DA enrichment indicates that the CB–DA composite achieved a 90% capture efficiency within 3 min at an *E. coli* concentration of 10^6^ CFU/mL. Furthermore, SEM images of HCT-116 cells bound to the CB–DA surface are shown in Figure 4C. These experiments confirm that CB–DA is useful for the biocompatible enrichment of pathogens and cells.

### 2.5. Nucleic Acid Isolation Using the Biocompatible Composite

To further confirm that the CB–DA composite rapidly and effectively adsorbed the bacteria, the fluorescence signals from real-time PCR analyses of amplified DNA extracted from the supernatant and precipitate (Figure 5A) of the *E. coli* (CFU 10^4^, 1 mL) enrichment using a Qiagen kit (100 μL of tested sample) are shown in Figure 5B. The inset figure shows the melting-curve plots, which represent the amplification products from the systems (black line: 10^4^, 100 μL as a positive control; red line: supernatant from the CB-DA-treated sample, 100 μL; blue line: enrichment with CB-DA, 100 μL of precipitate; and green line: distilled water (DW) as a negative control). As shown in Figure 5B, the cycle threshold (Ct) value was approximately two cycles earlier for pathogen enrichment by the CB–DA enrichment system than that of using the kit. According to RT-qPCR amplification theory, two cycles earlier corresponds to a template concentration that is four-fold higher [22]. In addition, we simply measured the amount of DNA (low molecular weight from salmon sperm, 31149-10G-F) captured by DA or CB–DA in 5 min. DNA (100 μL; 0.1 mg/mL) was added to 1 mL of DA (50 mg/mL) or CB–DA (50 mg/mL), and the surface charges of the DA–DNA and CB–DA–DNA mixtures after washing out the free DNA are shown in Figure 5C. The reduction in the surface charge of the CB–DA–DNA indicates that the CB–DA composite is more effective at capturing the nucleic acid, and this effect may be due to the previously reported covalent bonding, physical adsorption, electrostatic interactions, and heterogeneous surface binding intrinsic to supermolecular family members [23,24].

## 3. Experimental Section

### 3.1. Chemicals and Reagents

All of the reagents were of an analytical grade and were used without further purification. Ammonium hydroxide solution (28% NH_3_ in H_2_O, 99.99% trace metals basis), 3-aminopropyltriethoxysilane (APTES), and 3-aminopropyl(diethoxy)methylsilane 97% (APDMS) were purchased from Sigma-Aldrich (St. Louis, MO, USA). Cucurbit[6]uril hydrate (C_36_H_36_N_24_O_12_, 94544-1G-F) was also obtained from Sigma-Aldrich. Proteinase K solution (Mat. No. 1014023, Qiagen, Germany) is commonly used to digest proteins and remove contaminants in nucleic acid preparations. The QIAamp DNA buffer system was from Qiagen (Hilden, Germany). The biocompatible DE (powder) and sodium bicarbonate (NaHCO_3_) used in the nucleic acid extraction were purchased from Sigma-Aldrich. Milli-Q water, ethanol (95–100%), and phosphate-buffered saline (PBS) (10×, pH 7.4) (Thermo Fisher Scientific, Waltham, MA, USA) were used in all experiments.

### 3.2. Biological Samples

The eukaryotic cells (HCT-116 colorectal cancer cells) were maintained in plastic culture dishes in high-glucose Dulbecco’s Modified Eagle’s Medium (DMEM) (Life Technologies, Carlsbad, CA, USA) supplemented with 10% fetal calf serum in a 37 °C humidified incubator with 5% ambient CO_2_. After culturing, genomic DNA was extracted from the cells using a spin column-based kit (Qia-kit) and a nanocomposite method. Prokaryotic *E. coli* (ATCC 25922) cells were inoculated into either nutrient broth medium or Luria-Bertani medium and then incubated overnight at 37 °C with shaking. The primers for the downstream analyses of the eukaryotic and prokaryotic cell are listed in Table 1.

### 3.3. Preparation of Biocompatible Composites

The gravity-powered washing method was used to remove fragments from the commercial DE to prepare uniform DE. For the production of the amine-modified diatomaceous earth (DA), the modification efficiencies of two types of amino polydimethylsiloxanes were compared. The electrokinetic potentials of the composite surfaces were assessed by measuring the zeta potentials. Briefly, serial aliquots of APTES or APDMS were added dropwise into 1 mL aliquots of 95% ethanol solution, followed by manual shaking for 3 min at room temperature (RT). Subsequently, 50 mg of DE was added to the amino solution with stirring. The optimal reaction ratio and modification time were determined. The amino-functionalized DA was washed, collected by centrifugation, and dried in a vacuum overnight at RT. The DA powder was stored in a reagent bottle. Preparation of the CB–DA was performed via the microwave method. Briefly, 50 mg of DA was dissolved in 1 mL of DI water to form a 50 mg/mL DA solution. CB (25 mg) [6] was added to 1 mL of DI water, and this solution was then sonicated for 1 min using an ultrasonic instrument. Subsequently, 20 μL of the 25 mg/mL CB solution was added dropwise into 2 mL of prepared DA solution, followed by heating in a microwave oven for 1 min. The double-functionalized CB–DA was washed and collected by centrifugation and then dried in a vacuum overnight at RT. The CB–DA powder was stored in a reagent bottle.

### 3.4. Characterization of the Biocompatible Composites

The morphologies of the DE, DA, and CB–DA were characterized using field-emission scanning electron microscopy (FE-SEM) (JEOL JSM-7500F) to confirm a uniform size distribution and decoration of the DE with CB. The zeta potentials of the materials were acquired via dynamic light scattering (DLS) (DynaPro NanoStar, Wyatt, GA, USA). Fourier transform infrared spectroscopic analysis (FTIR) (JASCO 6300, JASCO, Easton, MD, USA) was performed on unmodified DE, DA, and CB–DA to obtain information about the chemical modification.

### 3.5. Cell and Pathogen Enrichment and Nucleic Acid Capturing

A Libra 22 UV/visible spectrophotometer was used to estimate the effectiveness of the composites for pathogen enrichment by measuring the absorbance of the cell and pathogen solutions at 600 nm. A vortex mixer (T5AL, 60 Hz, 30 W, 250 V) was used for mixing the sample wells in the enrichment system. A CF-5 centrifuge (100–240 Vas, 50/60 Hz, 8 W, 5500 rpm) was used for sample collection. QIAamp DNA/RNA Mini Kits (DNA Mini Kit: Cat No. 51304 and RNA Mini Kit: Cat No. 74104, Hilden, Germany) were used as spin column-based methods for isolating nucleic acids (following the manufacturer’s instructions) from both the supernatant and precipitate from the CB–DA enrichment system. An AriaMx real-time PCR system (Agilent Technologies, Santa Clara, CA, USA) was used to confirm and estimate the enrichment efficiency.

## 4. Conclusions

In this study, an efficient fabrication method for a supermolecular modified diatom composite (CB-DA) is presented, and this CB–DA composite was then used for efficient sample enrichment and in situ nucleic acid preparation from enriched prokaryotic and eukaryotic cells. The well-known amino functionalization technology was improved, as verified by surface charge characterization. APDMS modification, which involves diethyl groups, requires two sites for modification, while APTES modification, which involves triethyl groups, requires three sites for modification. Our results show that the latter approach leads to a higher density of amino modification. Our results also show that the supermolecule formation was maximized in the CB–DA composite, and that the encapsulation performance of the composite was improved via enhanced dispersion. We also showed that the novel CB–DA composite achieved a 90% capture efficiency within 3 min even at an *E. coli* concentration of only 10^6^ CFU/mL. We also observed that real-time PCR analysis of amplified DNA isolated using the CB–DA enrichment system showed a four-fold enhancement of the early signal, which might be useful for sample preparation to support early diagnosis. This study provides new insight into amino functionalization and lays a foundation for the further development of sample preparation techniques for human disease diagnostics and molecular encapsulation in drug delivery systems.

## Figures and Tables

**Figure 1 molecules-24-01321-f001:**
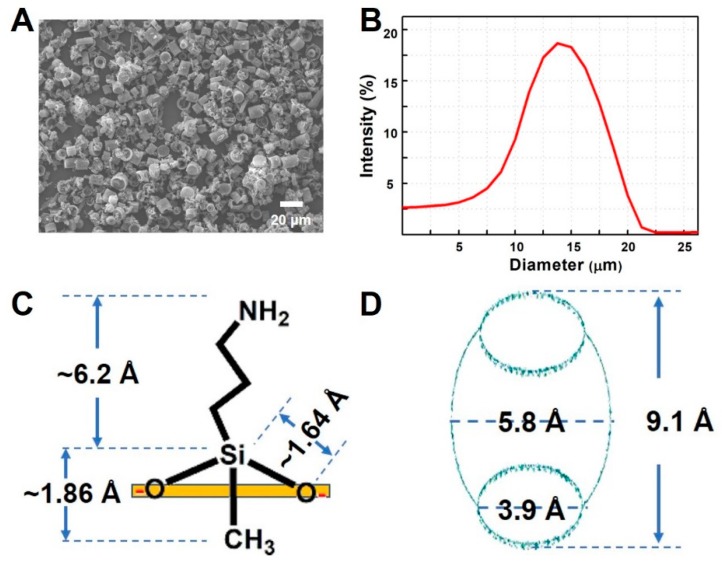
Characterization of the studied materials. (**A**) Scanning electron microscopy (SEM) image of diatomaceous earth (DE). (**B**) Dynamic light scattering (DLS) analysis of the size distribution of the DE. (**C**) Short chain length of amino organic compound (3-aminopropyl-methyl-diethoxysilane). (**D**) Diagram of cucurbit[6]uril, portal diameter (3.9 Å), cavity diameter (5.8 Å), and height (9.1 Å).

**Figure 2 molecules-24-01321-f002:**
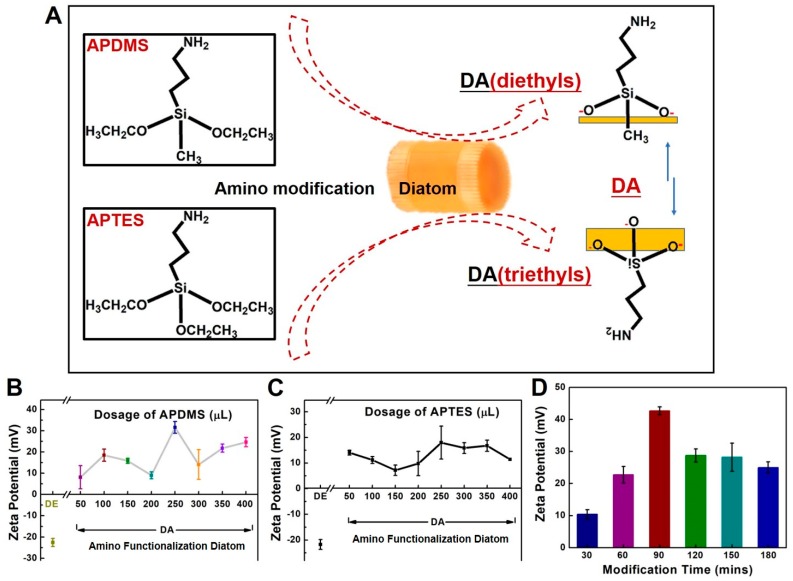
(**A**) Schematic of the process flow for amino functionalization of diatomaceous earth (DA). The diatom substrate was modified with either 3-aminopropyl-methyl-diethoxysilane (APDMS) or 3-aminopropyl-triethoxysilane (APTES). (**B**,**C**) Optimized conditions for amino functionalization of DE via APDMS (diethyls) and APTES (triethyls). Doses (μL) of APDMS and APTES reacted with 500 μL, 50 mg. L^−1^ DE in 1.5 mL tubes for 120 min. (The volume ratios ranged from 1/11 to 8/18.) APDMS and APTES (50 μL to 400 μL) were tested serially. (**D**) Optimization of the APDMS modification time. The zeta potentials of DA products with different APDMS modification times ranging from 30 to 180 min.

**Figure 3 molecules-24-01321-f003:**
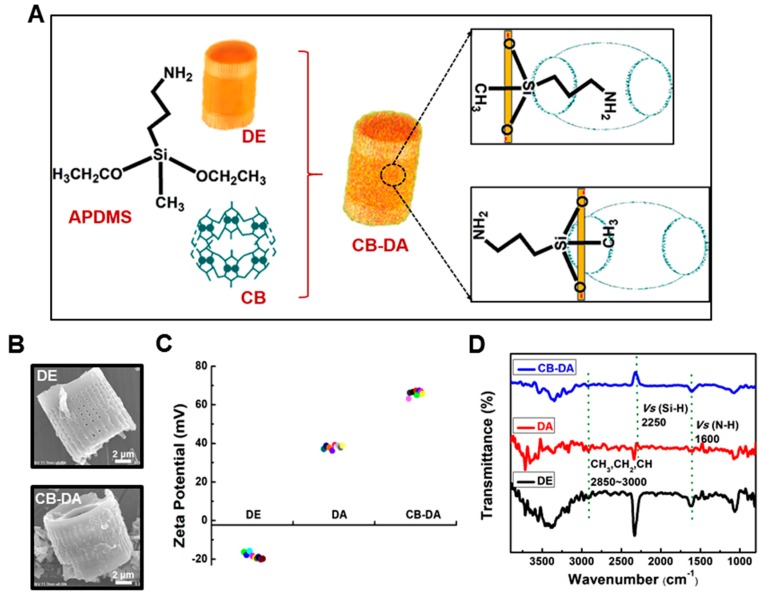
(**A**) Schematic representation of the cucurbituril modification of the amino-functionalized diatomaceous earth (CB-DA). The two free chains of APDMS on the substrate are encapsulated in the cavity of the cucurbit[6]uril (CB). (**B**) Scanning electron microscopy (SEM) images of DE and CB-DA. (**C**) Zeta potentials of the prepared materials: pure DE, amino-functionalized diatomaceous earth (DA), and cucurbituril-modified amino-functionalized diatomaceous earth (CB-DA). (**D**) Fourier-transform infrared (FTIR) spectrum analysis of the materials with dye. Pure DE (DE, black line), amine-modified DE (DA, red line), cucurbituril-coated amine-modified DE (CB-DA, blue line).

**Figure 4 molecules-24-01321-f004:**
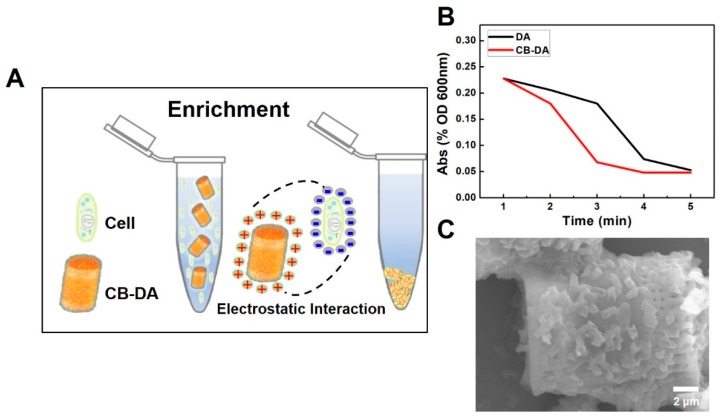
Pathogen enrichment schematic and demonstration. (**A**) Enrichment schematic; the electrostatic interaction between the positive surface of the CB–DA and the negative charge from the cell membrane. (**B**) The supernatant absorbances of the tested pathogen samples after DA and CB–DA *E. coli* enrichment (CFU 10^6^, 2 mL). (**C**) Cell enrichment demonstration. SEM images of the HCT-116 cells adhered to the surface of CB-DA.

**Figure 5 molecules-24-01321-f005:**
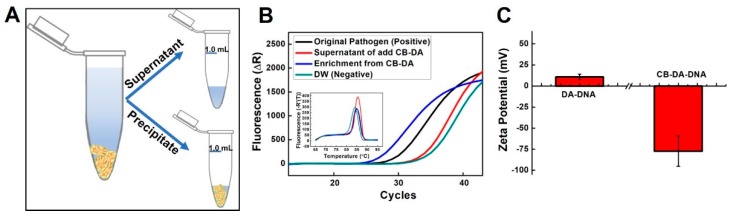
(**A**) A diagram of the supernatant and precipitate from the enrichment system. (**B**) Fluorescence signals from real-time PCR analyses of amplified DNAs extracted from the supernatant and precipitate following *E. coli* enrichment (CFU 10^4^, 1 mL) using a Qiagen kit (100 μL of the tested sample). The inset figure shows the melting-curve plots representing the amplification products from the systems. Black line—10^4^ CFU in 100 μL as a positive control; red line—supernatant with CB–DA addition, 100 μL; blue line—enrichment from CB–DA (100 μL of precipitate); and green line—distilled water (DW) as a negative control. (**C**) Zeta potential-based comparison of the nucleic acid capture efficiencies of the composites, i.e., DA-DNA and CB-DA-DNA.

**Table 1 molecules-24-01321-t001:** Primers used in this study.

	Primer	Sequences (5′→3′)	Annealing Temp. (°C)
*E coli*	*Ecoli-rodA-195F*	GCA AAC CAC CTT TGG TCG	58
	*Ecoli-rodA-195R*	CTG TGG GTG TGG ATT GAC AT	

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
