# Peer review of "A Sample Preparation Technique Using Biocompatible Composites for Biomedical Applications"

_molecules, 2019, doi:10.3390/molecules24071321_

Reviewer 1 Report

The author developed a sample preparation technique for biomedical application, such as PCR-based diagnosis application. For the work, I still have some questions:

1) in Figure A, please check the resolution that I can not distinguish error bar in the Figure 1A; For the DLS analysis, also, please generate error bar (figure 1B)

2)in Figure 2B, the zeta potential increased when treating with 200 uL, and achieved maximum at 250uL, but decreased a lot, any explanation about that? And in Figure 2C, regardless the minimum value at 200uL and maximum value at 250 uL. the zeta potential almost the same at a range of 200-400 uL, any explanation?

3) in Figure 2D, what is the concentration you used in the experiment to measure optimization of the APDMS modification time?

4) in the part 2.5 and Figure 5B, you showed the CB-DA enrichment system was 100-fold more effective for pathogen enrichment compared with original pathogen. Why? and which timepoint for calculate 100-fold? what is the accurate number of fluorescence in experimental and control groups?what is the centrifuge rate? 

Author Response

The author developed a sample preparation technique for biomedical application, such as PCR-based diagnosis application. For the work, I still have some questions:

1) in Figure A, please check the resolution that I can not distinguish error bar in the Figure 1A; For the DLS analysis, also, please generate error bar (figure 1B)

Ø  We thank the reviewer for the comment. We have redrawn the error bar in Figure 1A. And, the auto-correlation curve of the diatom has been shown in diameter-intensity which could show the size density into the curve area without error bar. Combined with the SEM image, we determined that the DE was well distributed around 10 to 20 μm in Figure 1B.

2) in Figure 2B, the zeta potential increased when treating with 200 uL, and achieved maximum at 250uL, but decreased a lot, any explanation about that? And in Figure 2C, regardless the minimum value at 200uL and maximum value at 250 uL. the zeta potential almost the same at a range of 200-400 uL, any explanation?

Ø  We thank the reviewer for the comment. As other studies reported that ‘The maxima could be due to the competing effects of increased surface area and the oxidation rate of the surface. Initial increase could be due to the increased area and the subsequent decrease could be due to the increased oxidation rate of the surface [1, 2, 3]. During our multiple experiments, the modification ratio at 1:2 (250 μL into 500 μL DE) showing good activity in both APDMS and APTES modification which may cause by the appropriate ratio of the surface area and aminosilane density (including aminosilane hydrolysis). And the density of aminosilane was plateaued out at range of 200~400 uL in APTES system which may cause by the saturated aminosilane.

3) in Figure 2D, what is the concentration you used in the experiment to measure optimization of the APDMS modification time?

Ø  We thank the reviewer for the comment. We have added detailed explanation in the manuscript. Here, the modification ratio at 1:2 (250 μL into 500 μL DE) was studied to measure optimization of the APDMS modification time. Notably, we referred the reported APTES modification procedure, 120 minutes incubation (modification) time has been fixed on the dosage studies. Based on the time-optimized test, the modification ratio at 1:2 (250 μL into 500 μL DE) in 90 minutes has been chosen for the APDMS incubation in this study. 

4) in the part 2.5 and Figure 5B, you showed the CB-DA enrichment system was 100-fold more effective for pathogen enrichment compared with original pathogen. Why? and which timepoint for calculate 100-fold? what is the accurate number of fluorescence in experimental and control groups? what is the centrifuge rate? 

Ø  We thank the reviewer for the comment. It was a mistake for calculation. As shown in Figure 5B, the Ct value was approximate 2 cycles earlier for pathogen enrichment by CB-DA enrichment system than that of using the kit. According to RT-qPCR amplification theory, 2 cycles earlier corresponds to a template concentration that is 4-fold higher [22].

Ø  The centrifuge was used in this study to spin-down the precipitation simply is a small-convenient device (CF-5, 5500 rpm).

Reviewer 2 Report

In this paper, Shin et al. present a novel method to absorb pathogens using amine modified composites. This work is based on their recent study published on ACS Sustainable Chemistry and Engineering in 2019. The author replaced the APTES with APDMS for more efficient modification. The authors optimized the functionalization condition, showed that cucurbituril-diatomaceous earth complex could enrich E. coli and HCT-116 via electrostatic interaction thus implied its biomedical application. The authors provide a complete and convincing story which will attract the general interest of the readership.

I have some concern on this manuscript:

Major concern:

As the author improved the modification efficient by using APDMS, more comparison besides the Zeta potential should be given, e.g. Abs at 600 and rtPCR. With APDMS, would the enrichment be more efficient? Even for Zeta potential, the difference is vague (Fig 2 B and C). The error bar at 300 uL in Fig 2B is large and the reason for the potential peak at 250 uL is not given.  Can it be an artifact?

Minor concerns:

1. Fig 1A and Fig 4C, the scale bar cannot be read at all. It should be drawn larger.

2. Line 92, “…….APDMS and APTES, respectively, led to differences in the surface modifications resulting from the use of these two materials.” This is wordy and ambiguous.

3. Fig 4B, the abs at 600 nm decreased from 0.24 to 0.05. However, the linear absorbance range of most spectrometers is between 0.1 and 1, so one cannot conclude that “CB-DA composite achieved a 90% capture efficiency” Start at abs higher than 0.24 if possible.

Author Response

In this paper, Shin et al. present a novel method to absorb pathogens using amine modified composites. This work is based on their recent study published on ACS Sustainable Chemistry and Engineering in 2019. The author replaced the APTES with APDMS for more efficient modification. The authors optimized the functionalization condition, showed that cucurbituril-diatomaceous earth complex could enrich E. coli and HCT-116 via electrostatic interaction thus implied its biomedical application. The authors provide a complete and convincing story which will attract the general interest of the readership.

I have some concern on this manuscript:

Major concern:

As the author improved the modification efficient by using APDMS, more comparison besides the Zeta potential should be given, e.g. Abs at 600 and rtPCR. With APDMS, would the enrichment be more efficient? Even for Zeta potential, the difference is vague (Fig 2 B and C). The error bar at 300 uL in Fig 2B is large and the reason for the potential peak at 250 uL is not given.  Can it be an artifact?

Ø  We thank the reviewer for the comment. The APDMS related study has been reported in our previous study [4]. We had studied the chemical characterization (Electron spectroscopy for chemical analysis (ESCA) spectra) and efficiency of its application, but here we expected to study the possible mechanism (electrostatic interaction). Therefore, we focused on the zeta potential of the composites in this study.

Ø  As other studies reported that ‘The maxima could be due to the competing effects of increased surface area and the oxidation of the surface. Initial increase could be due to the increased area and the subsequent decrease could be due to the increased oxidation of the surface [1, 2, 3]. During our multiple experiments, the modification ratio at 1:2 (250 μL into 500 μL DE) showing good activity in both APDMS and APTES modification which may cause by the appropriate ratio of the surface area and aminosilane density (including aminosilane hydrolysis).

Minor concerns:

1. Fig 1A and Fig 4C, the scale bar cannot be read at all. It should be drawn larger.

Ø  We thank the reviewer for the comment. We have redrawn the error bar in Fig 1A and Fig 4C.

2. Line 92, “…APDMS and APTES, respectively, led to differences in the surface modifications resulting from the use of these two materials.” This is wordy and ambiguous.

Ø  We thank the reviewer for the comment. We have rephrased this sentence “The diethyls of APDMS is differences from the triethyls of APTES for the surface modification of materials. Two points of the bonding site from the APDMS is expected to be more efficiency on bonding than APTES with three points (Figure 2A)”.

3. Fig 4B, the abs at 600 nm decreased from 0.24 to 0.05. However, the linear absorbance range of most spectrometers is between 0.1 and 1, so one cannot conclude that “CB-DA composite achieved a 90% capture efficiency” Start at abs higher than 0.24 if possible.

Ø  We thank the reviewer for the comment. We had reported the pathogen capturing ability of CB-DA composites in the water treatment test with a high absorbance range [3]. Here, in order to study the CB-DA composites in sensitive diagnostics which should face with the pathogenic bacteria in complex matrices at very low concentrations, we focused on the enrichment efficiency of CB-DA with low concentration. Therefore, the studied absorbance range was start at 0.24.